# Technologies for the Rational Use of Animal Waste: A Review

**Ruslan Iskakov** [1],* and **Adilet Sugirbay** [1,2]

1. Technical Faculty, S. Seifullin Kazakh Agrotechnical University, Astana 010000, Kazakhstan
2. College of Mechanical and Electronic Engineering, Northwest A&F University, Xianyang 712100, China
* Correspondence: r.iskakov@kazatu.kz; Tel.: +77-05-116-4648

**Abstract:** Animal waste can serve as a raw material source for feed preparation, and can also be used, after appropriate processing, as fuel, fertilizer, biogas, and other useful products. In addition, the practical use of these wastes eliminates their mandatory disposal. Recycling animal waste is a feature of the circular economy, leading to environmental sustainability. In this regard, we conducted a search and review of contemporary scientific publications from open sources, including publications and data from Internet portals, Web of Science, Scopus scientometric databases, websites of patent offices, libraries, and reading rooms. It has been found that animal by-products are desirable for use in combination with vegetable protein sources. The 15 most relevant types of animal waste and their use are indicated based on current scientific publications. Moreover, 13 types of feed of animal origin, along with their purposes and descriptions, are also identified. Current scientific publications and research on the processing of insects into feed; the use of bird droppings, meat, and bone and bone meal; and the processing of seafood waste, bird waste, and eggshells are reviewed. As a result, firstly, the most important types of technological equipment involved in animal waste processing technologies, particularly devices for drying, grinding, and mixing, are analyzed and discussed. Secondly, technologies for processing waste into useful products of animal origin are analyzed and discussed.

**Keywords:** animal waste; feed of animal origin; processing; drying; grinding; mixing of feed particles; waste-free technologies

## 1. Introduction

By 2050, the world's population is estimated to reach 9.9 billion people, which, along with increased urbanization and rising incomes, will increase demand for livestock products. Global demand for livestock feed will almost double [1]. With such a high demand for feed, an increase in feed production will be necessary. An increase in feed production can be achieved through the processing of agricultural and food waste [2], as well as the development and use of efficient and intensive technology. The production of agricultural waste, such as waste from crop production, animal husbandry, and the food industry, will reach 998 million tons per year [3]. Adequate use of animal waste reduces the amount of such agricultural waste and its negative effect on the environment [4]. Furthermore, managing animal waste is a feature of the circular economy, leading to environmental sustainability [5].

Feeds of animal origin include poultry by-product meal, fish paste and fish meal, bone meal, bone and meat meal, chicken by-product meal, overcooked chicken innards, egg waste, hydrolyzed feather meal, intestinal mucosal meal, blood meal, high protein whey powder, and insect meal. Animal ingredients can be used as the sole source of dietary protein or in combination with additional vegetable protein sources [6]. At the same time, animal by-products must be used following the rules of the European Union (EU), including in feed [7–10]. Eliminating the use of animal products in the production of feed can lead to severe environmental, economic, and social impacts, such as waste of slaughterhouse resources, pollution of the natural ecosystem, and the cost of feed production due to the demand for higher value-added components [11]. For example, one

of the popular methods used to remove animal waste products is rendering. Carcass meal (protein powder), tallow, and water are produced by the conversion of animal carcasses or wastes. This is accomplished through mechanical, thermal, and chemical processes. The mechanical processes include grinding, mixing, pressing, decanting, and separating. The thermal processes include boiling, evaporation, and drying. The chemical process includes solvent extraction. During rendering, animal excrement is dried and fat is separated from the bones and proteins. The resulting fat is used as a cheap raw material for the production of fat, animal feed, soap, candles, and biodiesel [12]. At the same time, the importance of improving non-waste technologies for processing a wide range of available waste raw materials, as well as developing innovative methods for the production of useful products from waste raw materials, should be noted. Therefore, this line of research is of great scientific and practical importance, and is relevant for analysis and generalization.

The purpose of this study is to summarize scientific data in the field of technologies for the rational use of animal waste following the principles of the circular economy. To achieve this goal, the following research objectives were set:

- Analyze and discuss the most important types of technological equipment involved in animal waste processing technologies, particularly devices for drying, grinding, and mixing;
- Analyze and discuss technologies for processing waste into useful products of animal origin.

Based on the foregoing, the highest level of attention was paid to research on the processing of waste generated by meat processing plants and poultry processing factories, as well as public catering facilities. At the same time, scientists pay great attention to the improvement of technological equipment, with the help of which it is possible to carry out mass processing of waste, thereby obtaining various useful products by means of existing technologies. It is expedient to process complex raw materials with the maximum beneficial effect according to the requirements of the circular economy. Based on production experience and safety requirements, researchers pay attention to the disinfection of waste raw materials of animal origin. In this regard, during processing, high-temperature processing is used in the process of cooking and/or drying, with the effect of sterilization in some cases, depending on the specific animal waste. In addition, attention is drawn to the phased grinding (preliminary and final), the duration of exposure to drying agents, the sorting of waste raw materials, the use of heat treatment methods, and the combination of technological processes in one apparatus. Considerable attention is paid to the intensification of processes in technological lines, which implies obtaining high technical and economic indicators, using efficient working bodies, eliminating dangerous zones, and employing multifunctional devices. At the same time, due attention should be paid to the effectiveness of methods for designing and constructing parts and assemblies of process equipment. At the same time, active scientific research is being carried out to develop theoretical substantiations of scientific hypotheses.

Work is underway to model and simulate all kinds of situations to put into practice the most optimal technologies for processing animal waste. Many scientific developments are based on alternative proposals, and comparative characteristics are based on the demands of time and customers. Research and experiments are being actively conducted on the digestibility of feed from various raw materials of animal origin by farm animals, birds, and domestic animals. Many researchers are conducting scientific work to find new and effective ways to use the generated waste of animal origin for practical use in life. Considerable attention is being paid to the search for new sources of raw materials for feed production. For example, insect feed flour is used. Insects are considered an alternative to fishmeal. Waste of raw animal materials of aquatic origin is widely used for the preparation of feed. In many countries, processed animal by-products are used as soil fertilizer, biogas, biochar, biofuel, bio-oil, feed, soap, glue, and other useful products. Based on the importance of this area of research, scientists are conducting active scientific work in the field of the circular economy and suggesting evidence-based developments that should be summarized and analyzed.

## 2. Materials and Methods

This review article contains an analysis of data from open sources, including publications and data from Internet portals, Web of Science, Scopus scientometric databases, websites of patent offices, libraries, and reading rooms. In the process of analyzing the scientific, technical, and patent-licensed literature, the following were considered: trends in the development of devices that combine the processes of drying, grinding, and mixing of particles of animal origin in the preparation of feed; global animal waste markets; assessment of the dynamics of the development of device designs for such important processes as drying, grinding, and mixing of feed meal particles from animal waste; the key developers of drying–grinding and drying–mixing devices; and the key segments and technologies for the use of waste raw materials of animal origin, especially to obtain feed. In writing this traditional review, a total of 200 papers were analyzed on the development of devices for drying, grinding, and mixing animal feed particles, types of uses for animal waste, and all kinds of animal feed. The criteria for inclusion of materials in this review were studies related to the use of simultaneous drying and grinding in one apparatus for the production of feed flour from animal waste, mainly for the period 2012–2022; studies on the use of animal waste for feed preparation; and studies on complex processing of raw materials of animal origin to obtain dry animal feed. The keywords used in the search for scientific and technical information were the following: drying with grinding, bone and meat and bone feed meal, dryers–grinders, the combination of drying with grinding and mixing, dry animal feed, and complex processing of raw materials of animal origin.

## 3. Results and Discussion

### 3.1. Devices for Drying, Grinding, and Mixing in the Processing of Animal Waste

Of particular importance in increasing the production of feed flour is the use of all types of non-food raw materials, waste, and confiscated goods from meat processing enterprises and farms [13,14]. At the same time, various methods for the production of feed flour from animal-origin waste raw materials are widely used, and new devices are being developed to carry out the necessary processes for processing waste of animal origin.

A known method for the production of meat and bone meal includes the processing of animals confiscated by veterinarians, dead animals, birds, slaughterhouse waste, and other meat and bone waste. The aim was to develop small-sized equipment that would guarantee the production of a high-quality feed product of high biological value. However, this development has not found wide application [15].

The studies carried out by the Ukrainian Research Institute of the Meat and Dairy Industry made it possible to establish the feasibility of using the method of combined high-temperature drying of defatted greaves in a suspended state with simultaneous grinding, which makes it possible to significantly intensify the drying process and improve the quality of the resulting feed [16,17].

It was proposed that a unit representing a two-section cylindrical-conical dryer with fluidized layers in sections be used: receiving (drying in a thin "fluidized bed") and drying (drying in a spouting "fluidized bed"). This was performed in the Research Institute named after V.M. Gorbatov [18]. At the same time, studies have been carried out in the field of heat and mass transfer, on topics such as the creation of vibrocutting plants for grinding meat and cutting tubular bones; the creation of resource-saving technologies for processing by-products, including bone, raw fat, and blood; and the creation of equipment complexes for their implementation. The Ya8-FLK bone processing line was put into mass production. Technology has been created for the complex processing of tubular bones of cattle for the production of edible fat, ornamental bone, and fodder broth. Its application allowed for a significant reduction in the energy consumption caused by the process of drying defatted and polished bone [19,20].

Known pneumatic convective dryer company "Stork-Duc" (Holland) is designed for drying waste containing small amounts of fat, blood, feathers, wool, and more. Drying is carried out in a rectangular drying channel, into which the initial product enters from a screw feeder, and air is heated in a heater to 385–470 °C and fed through the grinder [21].

At the Shoket plant (Israel) and the Yekaterinburgsky Meat Processing Plant, an efficient technology for the production of high-quality feed flour from non-food waste was proposed and introduced into production [22].

A known drying–grinding plant [23] can be used for the manufacturing of feed flour. The installation includes housing in which a drying chamber and a mixing device equipped with blades, grinding elements, heat exchange chambers, a loading device, an unloading chamber, and a pre-drying chamber are located. The pre-drying chamber includes a cutter with an opening, as well as a mixing device made in the form of a rotor and equipped with blades.

The device for grinding and drying raw materials in the production of meat and bone meal includes a cylindrical housing with transverse partitions dividing it into sections; main and additional pipes for supplying raw materials; a drying agent; connecting pipelines; a shaft located in the housing with fixed disks and beaters; and elements to prevent the entrainment of unground particles. The purpose of this invention is to reduce energy losses and increase the productivity and quality of the finished product [24].

There is an invention (Figure 1) [25], similar to a grinding–mixing structure for food waste, in which the lower surface is a W-shaped curved surface. At the same time, the lower central part is a stirrer, and there is a heating block located on the outer surface for heating the inside of the vessel and transferring heat for the purpose of drying food waste. In the upper part, food waste is crushed.

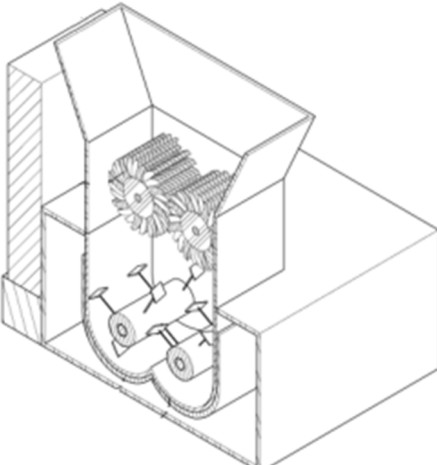

**Figure 1.** The part of the grinding device that mixes food waste. Retrieved from [25].

It is noted that the combination of grinding and its transportation, due to the forces of inertia rather than pneumatic transport, allows for a reduction in energy costs compared to the most common hammer mills [26]. In this case, it should be noted that impact failures of the rotary machine lead to amplitude modulation of the vibration signal [27].

Considerable attention should be paid to effective methods of design, construction, and calculation [28,29]. The calculation method of maximum collision forces in kinematic chains in virtue of their momenta is presented [30]. The simplicity of this method is considered an advantage due to algebraic equations. Collisions between animal waste and the hammer inside the hammer mill are applied as an example to demonstrate the suggested method. This method can be used by engineers while designing the equipment to measure the loads acting on certain parts during collisions, as well as to find the optimal geometric parameters of the equipment. The degree of crushing depends on the mass of the row of hammers, the radius of the center of mass of the hammer, the length of the working surface of the hammer, and the radius of the axes of the rotor suspension. With this in mind, the area and average diameter of the particles of crushed raw materials, the angular velocity of rotation of the rotor, the modulus of elasticity of the crushed material, and the operating coefficient were determined [31].

In many cases, feed production is carried out in production lines. Figure 2 shows the technological scheme for the production of fishmeal [32]. Here, the importance of consistent installation of technological equipment should be noted, taking into account the production of several types of useful products, including fat and feed.

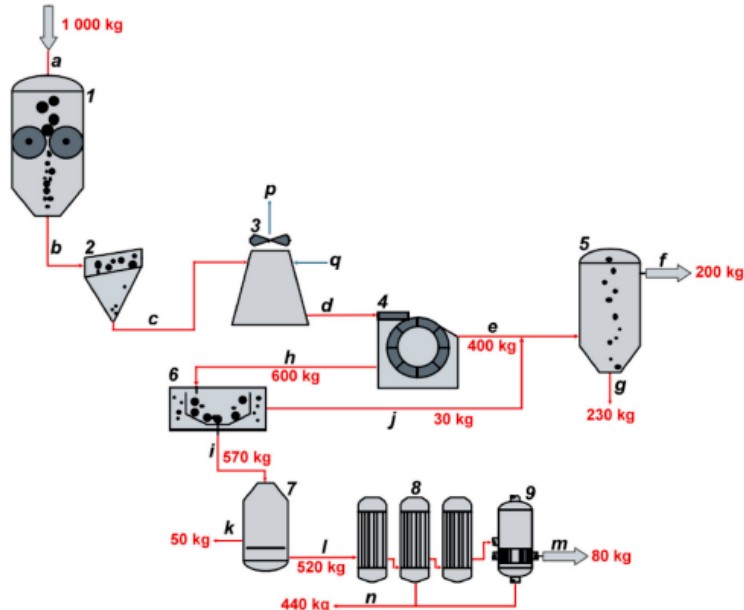

**Figure 2.** Technological scheme describing the line: a. Raw fish; b. mauled tissue; c. sieved gum; p. exhaust steam outlet; q. heat supply; d. cooked material; e. press cake; f. fish meal; g. waste vapors; h. stick water; i. fish water; j. sludge; k. fish oil; l. protein water; m. fish soluble paste; n. waste gasses; 1. grinder; 2. screen; 3. cooker; 4. screw press; 5. dryer; 6. decanter; 7. centrifuge; 8. double-stage waste vapor evaporator; 9. steam vacuum evaporator for the production of fishmeal. Retrieved from [32].

The work in [33] presents a batch digester mathematical model where the temperature, pressure, and stirrer speed are considered as factors to optimize the cooking process, and studies the effect of these factors on the quality of fodder meal from poultry waste. The response surface method (RSM) was used as an algorithm to optimize the rendering process. The results of the experiment showed that the batch cooking process can be optimized by maximizing protein percentage and minimizing fat percentage, moisture content, energy consumption, and pollutant emissions when the temperature, cooking time, and stirrer speed were 140 °C, 45 min, and 20 rpm, respectively.

An interesting patent [34] described a situation in which feed is produced from chicken manure by thermal decomposition and combustion of chicken manure occurs in a processing oven without an oxygen supply. The method allows the processing and safe conversion of chicken manure into feed by obtaining ash resulting from thermal decomposition, which has a high calcium content in a sterile state. The apparatus is able to carry out the chicken manure feed production method automatically.

There is a known patent [35] in which a variation of the cyclic method of cultivating chickens and earthworms is applied. It differs in that it includes the following steps: creating a breeding ground for earthworms next to the chicken coop, in which there is a grid with a container for centralized collection of chicken manure; and preparation of feed for earthworms, which consists of sterilizing the collected chicken manure and evenly distributing it on the field for breeding earthworms. The collected chicken manure is sterilized, dried, and made into artificial earthworm feed with other raw materials, and then sprinkled evenly to grow earthworms in aquaculture fields. The chicken feed is then prepared by moving sifted earthworms directly into the chicken coop so that the chickens can eat them, or mixing earthworms with other raw materials. Developed earthworms can also be sold.

Another patent includes a method for the preparation of organic fertilizer using chicken manure suitable for feeding Hermitia illucens. It includes aerobic and anaerobic fermentation of pre-treated chicken manure particles, collection of larvae, and mixing manure [36]. The novelty lies in the preparation of organic fertilizer using chicken manure, which includes pre-treatment of the manure, placement of pre-treated chicken manure particles into the fermentation tank, agitation of the chicken manure while spraying, uniform mixing and compaction for aerobic fermentation, agitation of chicken manure after aerobic fermentation, spraying and adding biological enzyme microbial material, uniform mixing and compacting for anaerobic fermentation, removing chicken manure particles, dispersing, processing as feed for feeding Hermitia illucens in the feeder, feeding Hermitia illucens larva, collecting Hermitia illucens larva from the feeder, using a separating device to separate Hermitia illucens manure from mixed excrement at the bottom of the feeder, and mixing Hermitia illucens with manure and humic acid to prepare the organic fertilizer.

Another known patent aims to prepare feed composition for the purpose of regulating the sugar metabolism of mulberry fish, which includes pouring a mixture of chicken manure and starter into a fermenter and carrying out the fermentation process with aerobic aeration [37]. The preparation of the feed composition includes pouring chicken manure powder, along with a mixture of bran and vegetables, into a grinder and grinding; adding water; drawing out a chicken manure mixture; throwing a mixture of chicken manure and sourdough into a fermenter; mixing the fermented chicken manure, inosinic acid, guanilic acid, dried shrimp, and snail powder for a fish food mixture; adding the fish food mixture to the extruder to extrude it; and finally, adding the fish food mixture (30–50 parts of water by weight and 2–3 parts), at a mixer speed of 200–300 rpm.

A known feed drying system (Figure 3) [38] contains a drying cylinder with a feed opening. First, the feed drying system starts up, the heating wire heats up, and the first fan turns on. Blowing starts, the feed for drying is poured into the working chamber from the feed opening, and then the transfer device begins to rotate the drying cylinder. In this case, the feed will be in contact with the heating layer. By this process, the moisture inside will evaporate, and the first fan will force air into the working chamber to speed up the work. The airflow in the cavity further enhances the evaporation of moisture, and at the same time, the air injected by the first blower will pass through the heating layer. The electric heating wire fills the working chamber with hot air, which further improves the dewatering efficiency. Hot air not only dries the feed, but also has the function of sieving the feed. The heat generated by the heating wire is completely concentrated in the working cavity of the drying cylinder, which can make full use of the energy and improve the drying efficiency.

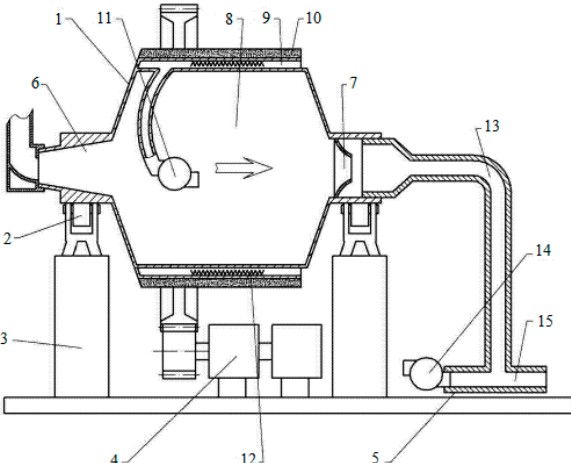

**Figure 3.** Feed drying system: 1. drying cylinder; 2. support roller; 3. base; 4. transmission device; 5. vacuum device; 6. material port; 7. outlet port; 8. working chamber; 9. heating layer; 10. insulating layer; 11. first fan; 12. electrical heating wire; 13. exhaust pipe; 14. second blower; 15. discharge pipe. Retrieved from [38].

A device for processing protein waste (Figure 4) [39] includes a mixing unit and a mobile grinding device installed on a movable platform, and contains a drying system. It is intended for the processing of protein waste used in the production of fuel for drying plants, as well as in animal feed or fertilizers.

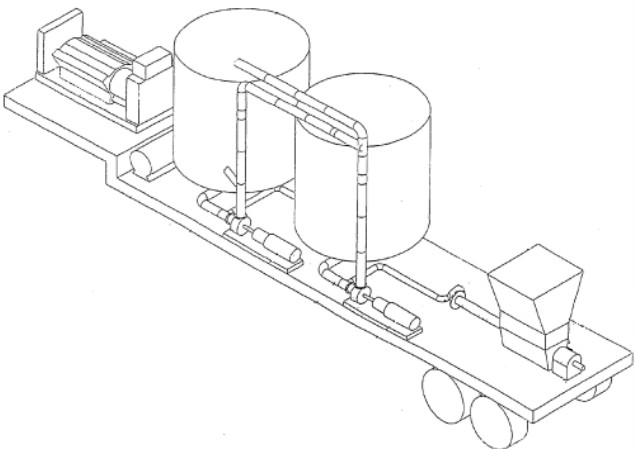

**Figure 4.** Device for processing protein waste. Retrieved from [39].

The known USI installation (Figure 5) [40] operates as follows. According to the given composition of the ingredients (fractions of fishmeal and waste of vegetable raw materials), the multi-component dispenser (13) forms the initial mixture for the mixer (14), which is fed into the body (4) of the thermal chamber through the loading hatch (6) with the electric motor (2) running. Inside the thermal chamber (4), the product reaches a blade (10) with grinding knives, mounted on the shaft of the rotor (9) of the mixing grinding device (8), which rotate, employing a connection with the drive (3) of the electric motor (2). The raw material is crushed while simultaneous being heated through the heat exchanger (5). For one USI operation cycle lasting 10–15 min, from 20 to 50 kg of raw material is loaded from the mixer 14. At the same time, soybean meal and/or bran and/or sunflower husks were used as waste plant materials. The grinding of raw materials in the heating chamber (4) utilizing the device (8) is carried out to a fraction size of 0.5–1.0 mm at a temperature of 120–150 °C, provided by heat exchanger (5). In this case, the steam from the heating chamber (4) is removed through the exhaust valve (11), and the moisture content of the finished product after the completion of the work cycle is 12–14%. Node (12) allows the user to control the operation of the USI and the characteristics of the drying–grinding process by starting and turning off the electric motor (2) to implement the specified work cycle, the temperature in the heating chamber (4), the size of the feed product fractions, the moisture content, and the mass of fractions being mixed in the mixer (14). When the feed products are ready, they are unloaded through a branch pipe (7) equipped with a damper valve. The installation additionally includes a multi-component dispenser for various types of raw materials, connected through a mixer with a heat chamber and made with the possibility of mixing fractions of the feedstock.

The development of extrusion technology has made it possible to propose new ways of recycling meat industry waste based on the dry extrusion method. Shredded waste of animal origin is pre-mixed with vegetable filler to reduce the moisture content of the mass fed into the extruder. The resulting mixture is subjected to extrusion processing, resulting in a product suitable for feeding. Grain, grain waste, bran, and meal can be used as filler. The volume of the filler exceeds the volume of waste of animal origin by 3–5 times and is determined by the moisture content of the waste. When the mixture passes through the compression diaphragms in the extruder barrel, the temperature and pressure increase due to friction. The time of passage of the mixture through the extruder does not exceed 30 s, and in the zone of maximum temperature it is only 6 s, so the negative effects of heat treatment are minimized. At the same time, during this time, the mixture is sterilized and

disinfected, increases in volume, homogenizes, stabilizes, and dehydrates. The extrusion technology for the utilization of biological waste, developed by Wenger Manufacturing, Inc. (USA), includes preliminary heat treatment of the mixture in the extruder conditioner, extrusion with steaming, and drying of the extrudate [41].

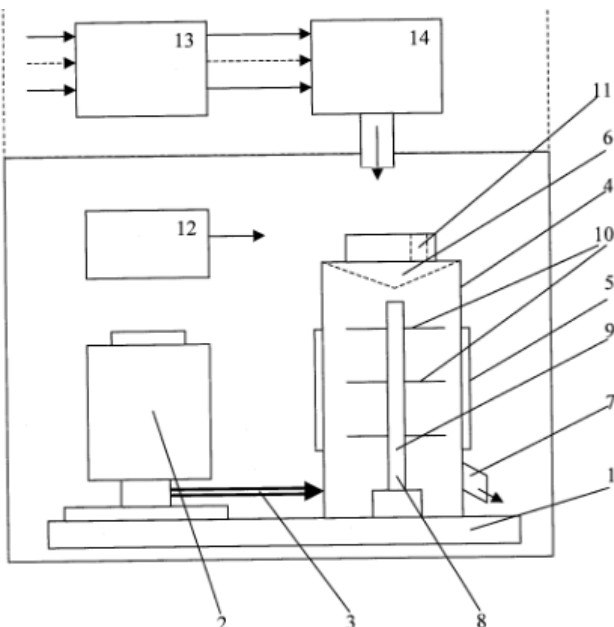

**Figure 5.** Installing the USI: 1. frame, 2. electric motor; 3. drive; 4. heat chamber housing; 5. heat exchanger; 6. loading hatch; 7. branch pipe; 8. mixing and grinding device; 9. rotor shaft; 10. blades; 11. outlet valve; 12. assembly; 13. multi-component dispenser; 14. mixer [40].

It is interesting to study different designs of mixing screws, the main difference between which is the pitch length. For screw designs, the calculated simulation results include screw powder feed rate, screw particle velocity, etc. [42].

There is an extrusion method for processing keratin-containing waste using an extruder that implements the principle of thermal vacuum exposure to raw materials (Figure 6) [43]. The extrusion process, pressing, grinding, and packaging are the main stages to process keratin-containing waste into feed flour. Firstly, the raw material is dried to a moisture content of 28% and cleaned of mechanical impurities to prepare for extrusion. Secondly, the keratin-containing waste is extruded and pushed into the barrel under the pressure of the screw press. In the extruder, the temperature rises by degrees, taking into account the increase in pressure. The feather changes from a solid to a melting state. The undigested keratin protein of the feather is transformed into a multispecies amino acid, 90% of which is composed of easily digestible crude protein. At last, the pen is ground into flour and packaged. The short-term processing of semi-dry raw materials at a temperature of more than 160 °C for 60–90 s is considered an advantage of this technology. Thermolabile amino acids in a physiologically digestible state are completely preserved during short-term hydrothermal treatment, and down-feather raw materials are sterilized.

To increase the productivity of the hammer mill, a combined sieve was proposed [44]. The effect of the combined sieve on the performance of a hammer mill was analyzed depending on the angle of attack of the material. When the mill used a round flat sieve, the pressure of the airflow field was found to increase from the center of the rotor to the end of the hammer using a simulation program. A layer of high-speed air circulation formed in the gap between the sieve and the hammer. The airflow field of the crushing chamber created a strong vortex motion, which made the airflow uneven and chaotic while the combination sieve hammer mill continued to operate. Moreover, this strong vortex motion constantly consumed energy, reducing the speed in the flow field and the pressure in the crushing chamber. The maximum pressure was 542 Pa. Compared to the round flat

sieve, the throughput increased by an average of 22.15%, and the electrical output per kWh increased by an average of 25.88%. The use of a combined sieve increased productivity and improved the over-grinding phenomenon, and the crushed particle size was comparatively uniform. In this case, it should be taken into account that for each hammer of a hammer mill, there is a random deviation. For the rows of hammers, chaos phenomena appear, with relative calm within a certain angle [45].

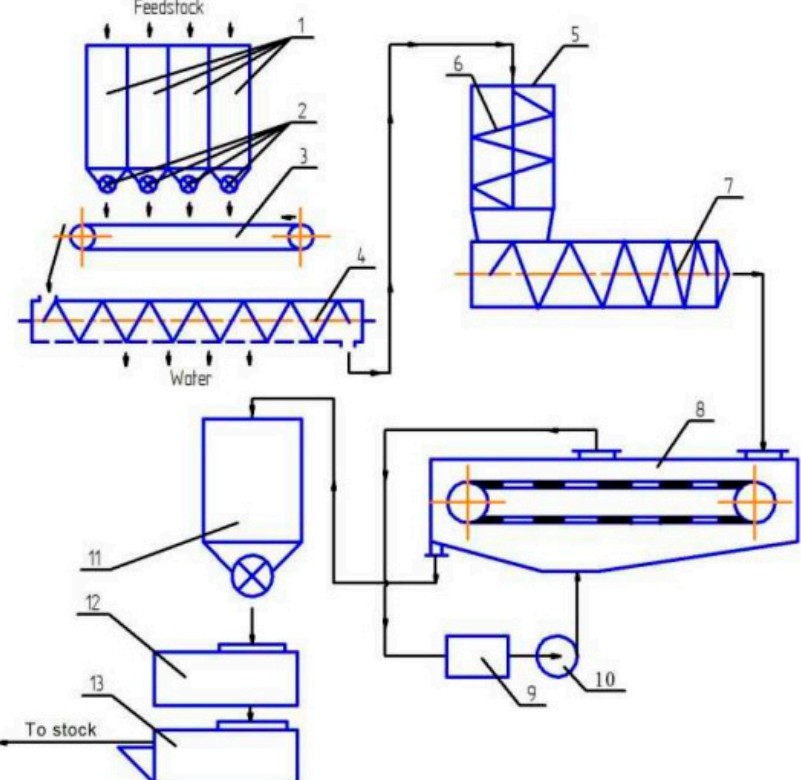

**Figure 6.** Technology for obtaining feed flour from keratin-containing waste by extrusion: 1. storage hopper; 2. dispenser; 3. screw conveyor; 4. moisture press; 5. screw extractor; 6. screw; 7. extruder; 8. dryers; 9. heater; 10. fan; 11. hopper; 12. chopper; 13. filling machine [43].

In the process of conducting scientific research, the following was revealed. As a result of the analysis of particle sizes, it was determined that the particle size decreases as the grinding rate increases [46]. It was found that the use of a fluidized bed in the process of convective drying intensified the technological process of drying particles [47]. Of no small importance is the use of optimal electric drives in the operation of devices for drying, grinding, and mixing [48].

Based on the foregoing, it should be noted that the nature of development work for devices for drying, grinding, and mixing depends on the nature of the enterprises, the level of unification of design conclusions, the level of knowledge of designers, the technical equipment and structure of the enterprise developer, and the required complexity and originality of the design and its development.

It should be noted that there are research results on the development of dryer–grinding apparatuses in a swirling flow [49–51], the development of impact elements for crushing [52,53], the development of technological lines [54,55] and separate devices for mixing [56,57], and devices for grinding [58], but the development of a combined, technical, and cost-effective device that simultaneously combines the technological processes of drying, grinding, and mixing of feed particles of animal origin into one apparatus has not been scientifically studied enough in the world.

*3.2. Technologies for Processing Waste into Useful Animal Products*

One of the common animal by-products is meat and bone meal used for domestic animals, birds, etc. In [59], the study aimed to investigate the degree of grinding of bone meal and meat, depending on their digestibility in the diets of dogs and the characteristics of feces. Meat and bone meal ground to various levels (MBM6, MBM12, and MBM24) was included in the main diet. Six beagle dogs (body weight: 16.7 kg $\pm$ 0.42) were used in the experiment for five days of feeding and five days of fecal collection. It was determined that the digestibility of organic matter, crude protein, and crude fat does not depend on the feed size of the particle. It was noted that different particle sizes of meat and bone meal did not significantly affect the concentration of fatty acids in the feces. These data from a pilot study suggest that the use of coarse or fine grinding, even with up to 24% meat and bone meal included in dogs' diets, does not adversely affect apparent protein digestibility or fecal quality.

Feeds rich in crude protein are associated with higher energy use [60] and are necessarily processed using mechanical and thermal processes—for example, in the production of nanometer rabbit bone meal. At the same time, the impact of various cooking conditions and cooking temperatures on the speed of sifting rabbit bone meal was emphasized. It was found that the effect of rotation speed on the particle dispersion index (PDI) was greater than the effect of time [61].

Animal waste is widely used in feed preparation and production of bio-seal, bio-coal, bio-oil, glue, and fertilizer [62]. The main types of animal waste and directions for their use are presented in Table 1. For example, Farmutili's development for heat (steam) generation makes it possible to use renewable energy from biofuels made from meal and bone meal instead of natural gas, which is burned in rotary kilns. The biofuel is exploited to heat the district heating system in Pyla city. The maximum heat generated from biofuels exceeds the need for district heating by 53.2% when burning 150,000 tons of meat and bone meal per year [63].

**Table 1.** Directions for the use of animal waste.

| Type of Animal Waste | Purpose of Animal Waste |
| --- | --- |
| Heads, skin, trimmings, fins, scales, entrails, and bones of fish [64] | Animal feed, biodiesel/biogas, diet foods (chitosan), food packaging (chitosan), cosmetics (collagen), and soil fertilizers [64] |
| Crustacean shells [64] | Animal feed, fertilizers, and for the removal of organic compounds from anionic metal compounds and heavy metal ions [65] |
| By-products of crab production [64] | Fish meal substitute [66] |
| Charru mussel, maçunim and oyster shell meals [67] | Source of calcium in the diet of European quails [67] |
| Processed chicken feathers (autoclaved with 0.5% NaOH) | The component in 2 mm granular granules intended for feeding fish [68] |
| Chicken fat | Feedstock for biodiesel production [69] |
| Bird meal | Biochar and bio-oil for use as heating and transport fuel [70] |
| Poultry slaughter waste | A ruminant feed with huge potential as cleaner animal feed for pollution prevention [71]; conversion to biogas by anaerobic digestion [72] |
| Bird droppings | Biochar and bio-oil for use as heating and transport fuel [70]. Raw material for fertilizer [73] |
| Broiler litter | Feed for male deer [74] |
| Bones of farm animals, bone food residues | Feed elements for creating diets for feeding pigs [75], glue [76] |
| Blood of farm animals | Feed blood meal [77] |
| Remains of bird feathers | Hydrolyzed feather meal [HFM] [77], bio-based glue [78] |
| Keratin waste from chickens, ducks, geese, turkeys (feathers, hair), goats, and sheep (wool, hooves, horns, nails) | Animal feed, fertilizers [79] |
| Egg shells, classified as waste by the food industry [79,80] | In the production of compound feed, in fish farming and feed for target animals, in the production of biochar [81], raw materials for growing Lingzhi mushrooms (Ganoderma lucidum) [82] |

The fishing industry is central to the economy for several countries in the world. More than 60% of fish and shellfish by-products are used as waste, including bones, trimmings, head, fins, skin, scales, and innards, while less than 40% of fish products are used for human consumption [66]. Fish skin offal is a source of collagen and gelatin. Fishbone mainly consists of cartilage mineralized with calcium phosphate. Fish spine waste is another source of protein and minerals. In addition, fish scales contain inorganic and organic components, mainly hydroxyapatite and collagen. Seafood by-products are mainly used as animal feed, biogas, food packaging (chitosan), dietary products (chitosan), soil fertilizers, and cosmetics (collagen). Saltwater fish offal is utilized as a substitute feed in pig diets to change conventional protein sources and satisfy protein requirements. The highly nutritious powder is processed from fish waste by drying and grinding. In the animal feed industry, crustacean shells have been recognized as a very good feed component. The heads, appendages, and exoskeletons of dried shrimp offal are notably rich in lysine. These types of animal waste contain various types of minerals [64]. It should be noted that feed elements are most often chosen to create the lowest-cost pig-feeding diets [77,78].

*Charru mussel*, *maçunim*, and *oyster shell* meals are exploited as sources of calcium [66]. Fishmeal, meat, and bonemeal from poultry waste are available sources of animal protein which are mainly used in Bangladesh when formulating poultry diets [83]. Table 2 presents the main types of feed of animal origin and their purpose.

**Table 2.** Types of feed of animal origin and their purpose.

| Type of Feed of Animal Origin | Type of Waste Raw Materials of Animal Origin | Purpose of Feed |
|---|---|---|
| Bone meal [61,76] | Feed elements for creating diets for feeding pigs [75]. Bones of agricultural animals, bone food remains [76]. Eggshell [84–86] | Broiler feed [83] |
| Meat and bone meal [77] | Heads, skin, trimmings, fins, scales, viscera, and bones of fish [64]; poultry slaughter waste | When compiling a diet for poultry [84] |
| Blood meal (dry hemoglobin meal, dried blood meal [DBM]) [77] | Blood of farm animals [77] | Fish food [85] |
| Fish feal [77] | Heads, skin, trimmings, fins, scales, entrails, and bones of fish [64] | When compiling the diet for poultry [84], in the diet of pigs [64] |
| Shrimp meal [64,77] | Dried shrimp production waste; consists of heads, appendages, and exoskeletons [64] | Broiler feed [64] |
| Chicken meal [77] (poultry meal) | By-products from butchering chickens (poultry) [77] | Broiler feed [78] |
| Pork meat meal [77] | Pig slaughter waste [77] | Food for orange-spotted grouper Epinephelus coioides [77] |
| Fermented feather meal [FFM] [77] | Remains of bird feathers [78] | Broiler feed [78] |
| Hydrolyzed feather meal [HFM] [77] | Keratin waste from chickens, ducks, geese, and turkeys (feathers, hair) [84] | Food for fish, including Cobia Rachycentron canadum, northern snakehead Channa argus, and halibut Scophthalmus maximus [77] |
| Feed for male deer [74] | Broiler litter [73] | Feed for male deer in the amount of up to 30% of broiler litter in the feed mixture [74] |
| Feed additives are processed into pellets along with other feed ingredients [87,88] | Cage laying litter, poultry litter [88], broiler litter [74] | Up to 30% of the cell droppings of laying hens in the feed mixture can be directed to Nellore lambs and kids [88]. Up to 30% broiler litter in the feed mixture can be fed to male deer [74] |
| Eggshell feed [89], compound feed component [90] | Waste of egg raw materials, including shells [90] | Feed for fish farming and target animals [90] |
| Insect meal, feed ingredient, substitute for fish meal (FM) and soy meal (SBM) in feed mixtures for poultry diets [91] | Insects of the orders Diptera (black lion fly, housefly), Coleoptera (mealworms), Megadrilacea (earthworm), Lepidoptera (silkworm and Ford Cyrinus), and Orthoptera (grasshoppers, locusts, and crickets) [91]. Black Soldier Fly Larvae (BSFL) [92–94] | Bird food [95] |

Various types of perishable organic waste and by-products are generated from poultry slaughter and post-processing. The yield of a bird carcass is usually more than 70% of the bird's live weight, and the rest is accepted as non-food waste, including bird blood, feathers, heads, paws, offal, and inedible entrails. Organic food waste management should focus on the production of aquaculture components or pet food [96]. It is known that processed chicken feathers are ground and then mixed. The mixed components are mixed with powdered additives satisfying the fish diet and then granulated into 2 mm granules [68]. It should be noted that chicken fat is a waste from poultry that is used to produce biodiesel [69,97]. With the help of thermochemical processes, biochar and bio-oil can be obtained from bird meal and bird droppings for use as heating and transport fuel [70]. The use of poultry slaughter waste in ruminant feeding has great potential as a cleaner animal feed product and to prevent environmental pollution [71].

It is known that drying bird droppings at high temperatures results in some loss of nitrogen. For example, Leto and Olsen found that during a quick drying process completed at high temperatures, there was a dependence between a quick increase in temperature and a loss of nitrogen [98]. Moisture inside bird droppings is the main cause of the unpleasant odor and rapid dissolution. The mixing of biomass ash with bird droppings allows us to reduce the moisture content, increase the nutrient content, and remove the odor and pathogens [73]. For example, the moisture content is reduced from 25.23% to 9.82% when 50% biomass ash is mixed with the poultry manure at 250 °C. The results show that more than 30% of broiler litter, without detrimental effects on efficiency, can be fed to male deer [74]. An experiment with three inexpensive feed additives was also formulated using caged laying litter and poultry litter pressed into pellets. Male Nellore lambs and local kids were evaluated using growth and metabolism tests. The type of concentrate addition did not affect the growth rate. Replacing caged laying litter and poultry litter did not significantly affect the digestibility ratios of the key nutrients and fiber fractions. All experimental animals had a positive balance of nitrogen, calcium, and phosphorus, meeting the requirements [88].

In [99], attention is drawn to the number of eggs with dirty shells or defects (broken, cracked, or without shell), as well as the weights and thicknesses of the shells. Feed production using spent eggs includes (a) collection of waste egg raw materials discarded after vaccine production from hatching eggs by medical companies; (b) mixing water and filler in feed materials with a moisture content of 25–40%; (c) material sterilization and cooling at a given temperature; (d) mixing the mixed material with beneficial microorganisms and fermenting at the temperature of cooling; (e) drying the fermented mixed material; and (f) grinding the dried mixed material according to the need for fish farming and feed for the target animals [90].

Insects can be a viable solution to the problem of reusing and adding value to food by-products. In the field of environment and economics, the ability of some insects to bioconvert waste into valuable products finds innovative applications [100]. Insects and black soldier fly larvae (BSFL) are considered potential substitutes for soybean meal and fish meal in feed mixtures as a source of protein and other active substances [91,92,101]. BSFL has the biological ability to convert the remaining energy of the previous waste into a new protein.

## 4. Conclusions

Contemporary scientific publications and research are mainly devoted to the processing of insects into feed; the use of bird droppings in the production of biochar, bio-oil, and fertilizers; and the preparation of feed additives. Meat and bone meal are mainly used for domestic animals and birds, but also find use, similarly to bone meal, in heat supply. Research is being carried out extensively on the processing of fish waste, crustacean shells, shrimp production waste, crab by-products, bird waste, and eggshells. The existing and emerging types of waste of animal origin direct researchers to think in depth about the rational processing of this source of raw materials to extract the most beneficial effect. At the same time, processing indicates the intensification and improvement of the quality of

such important processes as drying, grinding, and mixing, which are necessary for the production of feed flour, feed mixtures, fertilizers, biofuels, and other useful products. One of the best processing methods is the combination of drying processes with grinding and mixing in one apparatus, which can significantly increase the rate of heat and mass transfer, and reduce the cost of producing useful products due to the simultaneous processing of waste in one apparatus. At the same time, it should be noted that the addition of the mixing process will make it possible to prepare feed from a variety of raw materials of animal origin, which will eliminate the sorting process and reduce the duration of mixing due to the longitudinal-transverse, vortex movement of feed flour particles, which contributes to high uniformity.

Along with the fact that the combination of grinding–mixing with drying increases the uniformity of heat treatment and the distribution of product particles by fractions and composition, and also allows for sufficiently deep dehydration of the material, there are some disadvantages. In particular, these are the uneven distribution of processed particles in the temperature flow and the varying moisture content in the processed particles of raw materials and in parts of the apparatus. The latter, in some cases, causes a certain difference in the moisture content in the dried particles of raw materials, and in some cases causes overheating of finely crushed particles, along with simultaneous energy consumption and a decrease in the biological value of the product.

**Author Contributions:** Conceptualization, R.I. and A.S.; methodology, R.I.; software, A.S.; validation, R.I.; formal analysis, A.S.; investigation, A.S.; resources, A.S.; data curation, R.I.; writing—original draft preparation, A.S.; writing—review and editing, A.S.; visualization, R.I.; supervision, R.I.; project administration, R.I.; funding acquisition, R.I. All authors have read and agreed to the published version of the manuscript.

**Funding:** This research was funded by the Science Committee of the Ministry of Education and Science of the Republic of Kazakhstan, grant number AP09259673.

**Institutional Review Board Statement:** Not applicable.

**Informed Consent Statement:** Not applicable.

**Data Availability Statement:** Not applicable.

**Conflicts of Interest:** The authors declare no conflict of interest.

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
