# Peer review of "Technologies for the Rational Use of Animal Waste: A Review"

_sustainability, doi:10.3390/su15032278_

Round 1

Reviewer 1 Report

The review article “ Technologies for the rational use of animal waste: a review” is very interesting and I believe can contribute to the relevant literature. However, some revisions are required to be made to the manuscript before it can be recommended for publication. Please find my comments as follows.

1.      The abstract is confusing and does not read well. For instance, by reading “ In turn, obtaining meat and dairy products significantly depends on the supply of highly nutritious feed to animals”, I was thinking that the paper might be focused on using animal waste to feed other animals. Please rewrite the whole abstract and make clear what the purpose of the research is, what is the methodology applied, and what are the results obtained. In the current version, none of these points are clear.

2.      In the introduction section, please clarify the aims and objectives of the research. Also formulating the main research questions would clarify the direction of the review.

3.      The introduction section is too short. Please add some relevant points of the literature review to the introduction section to clarify the research background and highlight the gap that is covered int his research.

4.      Please add a paragraph to the end of the introduction section and explain the structure of the manuscript.

5.      In line 70, it is stated that “In writing this systematic review, a total of 200 papers were analyzed”. However, this manuscript is a traditional literature review and is not a systematic literature review (is there any search string formulated?). Please correct this part.

6.      The term “modern scientific research” has been used a few times throughout the text. However, “modern” is not an appropriate terms to address scientific research, please replace this term with a more proper one.

7.      I realized that the manuscript has been submitted to the special issue “Waste Management towards a Circular Economy Transition”. Although the content of the manuscript is in line with the special issue, I could not find anything in the manuscript pointing to circular economy. Please clarify (especially in the introduction and conclusion sections) how this study can help transitioning towards the circular economy. Since this manuscript is focused on animal waste, the authors can use the definition of circular economy linked with waste that is presented in the paper titled “Two decades of research on waste management in the circular economy: Insights from bibliometric, text mining, and content analyses” published in Journal of Cleaner Production to better highlight the role of managing animal waste in the transitioning towards the circular economy.

Author Response

Reviewer 1

The review article “ Technologies for the rational use of animal waste: a review” is very interesting and I believe can contribute to the relevant literature. However, some revisions are required to be made to the manuscript before it can be recommended for publication. Please find my comments as follows.

  1. The abstract is confusing and does not read well. For instance, by reading “ In turn, obtaining meat and dairy products significantly depends on the supply of highly nutritious feed to animals”, I was thinking that the paper might be focused on using animal waste to feed other animals. Please rewrite the whole abstract and make clear what the purpose of the research is, what is the methodology applied, and what are the results obtained. In the current version, none of these points are clear.

Response 1: Thank you for your comment! We have rewritten the abstract. Please, check!

Animal waste can serve as a raw material source for feed preparation, which can also be used after appropriate processing as fuel, fertilizers, biogas, and other useful products. In addition, the practical use of these wastes eliminates their mandatory disposal. Recycling animal waste is a feature of the circular economy leading to environmental sustainability. In this regard, we conducted a search and review of contemporary scientific publications from open sources, including publications and data from Internet portals, Web of Science, Scopus scientometric databases, websites of patent offices, libraries, and reading rooms. It has been found that animal by-products are desirable to be used in combination with vegetable protein sources. The 15 most relevant types of animal waste and their use are indicated based on current scientific publications. Moreover, 13 types of feed of animal origin, their purpose, and description are also identified. Current scientific publications and research on the processing of insects into feed, the use of bird droppings, meat and bone and bone meal, and the processing of seafood waste. animal origin, bird waste, and eggshells are reviewed. As a result, firstly, the most important types of technological equipment involved in animal waste processing technologies, particularly devices for drying, grinding, and mixing are analyzed and discussed. Secondly, technologies for processing waste into useful products of animal origin are analyzed and discussed.

  1. In the introduction section, please clarify the aims and objectives of the research. Also formulating the main research questions would clarify the direction of the review.

Response 2: (Lines 126-133)

The purpose of the study is to summarize scientific data in the field of technologies for the rational use of animal waste following the principles of the circular economy. To achieve this goal, the following research objectives were set:

- analyze and discuss the most important types of technological equipment involved in animal waste processing technologies, in particular devices for drying, grinding, and mixing;

- analyze and discuss technologies for processing waste into useful products of animal origin.

  1. The introduction section is too short. Please add some relevant points of the literature review to the introduction section to clarify the research background and highlight the gap that is covered in this research.

Response 3: (Lines 134-150)

Based on the foregoing, the main attention is paid to research on the processing of waste generated by meat processing plants and poultry processing factories, as well as public catering facilities. At the same time, scientists pay great attention to the improvement of technological equipment, with the help of which it is possible to carry out mass processing of waste, thereby obtaining various useful products under existing technologies. It is expedient to process complex raw materials with the maximum beneficial effect by the requirements of the circular economy. Based on production experience and safety requirements, researchers pay attention to the disinfection of waste raw materials of animal origin. In this regard, during processing, high-temperature processing is used in the process of cooking and/or drying, in some cases with the effect of sterilization depending on the specific animal waste. Also, attention is drawn to the phased grinding (preliminary and final), the duration of exposure to drying agents, the sorting of waste raw materials, the use of heat treatment methods, and the combination of technological processes in one apparatus. Considerable attention is paid to the intensification of processes in technological lines, which implies obtaining high technical and economic indicators, the use of efficient working bodies, the elimination of dangerous zones, and the use of multifunctional devices. At the same time, due attention should be paid to the effectiveness of methods for designing and constructing parts and assemblies of process equipment. At the same time, active scientific research is being carried out to develop theoretical substantiations of scientific hypotheses.

  1. Please add a paragraph to the end of the introduction section and explain the structure of the manuscript.

Response 4: (Lines 150-189)

Work is underway to model and simulate all kinds of situations to put into practice the most optimal technologies for processing animal waste. Many scientific developments are based on alternative proposals, and comparative characteristics based on the demands of time and customers. Research and experiments are being actively conducted on the digestibility of feed from various raw materials of animal origin by farm animals, birds, and domestic animals. Many researchers are conducting scientific work to find new effective ways to use the generated waste of animal origin for practical use in life. Considerable attention is paid to the search for new sources of raw materials for feed production. For example, insect feed flour is used. Insects are considered an alternative to fishmeal. Waste of animal raw materials of aquatic origin is widely used for the preparation of feed. In many countries, processed animal by-products are used as soil fertilizer, biogas, biochar, biofuel, bio-oil, feed, soap, glue, and other useful products. Based on the importance of this area of research, scientists are conducting active scientific work in the field of the circular economy and suggesting evidence-based developments that should be summarized and analyzed.

  1. In line 70, it is stated that “In writing this systematic review, a total of 200 papers were analyzed”. However, this manuscript is a traditional literature review and is not a systematic literature review (is there any search string formulated?). Please correct this part.

Response 5: Done Thank you for your comment! (Line 201)

In writing this traditional review, a total of 200 papers were analyzed on the development of devices for drying, grinding, and mixing animal feed particles, types of use of animal waste, and all kinds of animal feed.

  1. The term “modern scientific research” has been used a few times throughout the text. However, “modern” is not an appropriate terms to address scientific research, please replace this term with a more proper one.

Response 6: Thank you for your comment! We are using contemporary instead of modern.

  1. I realized that the manuscript has been submitted to the special issue “Waste Management towards a Circular Economy Transition”. Although the content of the manuscript is in line with the special issue, I could not find anything in the manuscript pointing to circular economy. Please clarify (especially in the introduction and conclusion sections) how this study can help transitioning towards the circular economy. Since this manuscript is focused on animal waste, the authors can use the definition of circular economy linked with waste that is presented in the paper titled “Two decades of research on waste management in the circular economy: Insights from bibliometric, text mining, and content analyses” published in Journal of Cleaner Production to better highlight the role of managing animal waste in the transitioning towards the circular economy.

Response 7: Thank you for your suggestion! Your suggestion was helpful! (Lines 1441-1443)

In the abstract: Recycling animal waste is a feature of the circular economy leading to environmental sustainability.

In the introduction: Furthermore, managing animal waste is a feature of the circular economy leading to environmental sustainability [5].

Dear reviewer, Thank you for your time and effort in reviewing our manuscript!

Reviewer 2 Report

It is a well-structured literature review, in which the authors focus on highlighting the 15 most relevant types of animal waste, and their use is indicated on the basis of modern scientific publications. Also, 13 types of feed of animal origin, their purpose, and description are also identified.

1. Authors are asked to indicate in the methodology section what were the inclusion or exclusion criteria of the papers to build the review.

2. In line 77, I consider it irrelevant to indicate that most of the publications are in English. Publications in other languages could be equally interesting and provide high quality, and it is well known that most of the communication of science and technology occurs in English.

3. In lines 96 - 100 it seems that the source of that information is missing. I suggest reviewing and citing the paper.

4. In this review, more information is needed regarding the composting processes of animal waste. I suggest reviewing this book or similar literature:

"Sánchez, Mercedes et al. (2008). Animal Solid Waste Management through Composting Techniques. Closed Semi-Continuous Composters as a New Approach for in-Situ Carcasses Disposal."

5. I suggest removing the term "As you know" throughout the document.

6. I suggest simplifying the conclusions, and focusing on the most relevant aspects found in the literature review.

Author Response

Reviewer 2

It is a well-structured literature review, in which the authors focus on highlighting the 15 most relevant types of animal waste, and their use is indicated on the basis of modern scientific publications. Also, 13 types of feed of animal origin, their purpose, and description are also identified.

  1. Authors are asked to indicate in the methodology section what were the inclusion or exclusion criteria of the papers to build the review.

Response 1. (Lines 201)

We have changed the word systematic review to traditional review.

In writing this traditional review, a total of 200 papers were analyzed on the development of devices for drying, grinding, and mixing animal feed particles, types of use of animal waste, and all kinds of animal feed.

  1. In line 77, I consider it irrelevant to indicate that most of the publications are in English. Publications in other languages could be equally interesting and provide high quality, and it is well known that most of the communication of science and technology occurs in English.

Response 2.

Thank you for your suggestion. We have deleted this sentence that indicated the publications were in English..

  1. In lines 96 - 100, it seems that the source of that information is missing. I suggest reviewing and citing the paper.

Response 3. (Lines 225-229)

Thank you for your suggestion. We have cited this piece of information.

The studies carried out by the Ukrainian Research Institute of the Meat and Dairy Industry made it possible to establish the feasibility of using the method of combined high-temperature drying of defatted greaves in a suspended state with its simultaneous grinding, which makes it possible to significantly intensify the drying process and improve the quality of the resulting feed [16-17].

  1. Yurchenko, P.A., Okranchuk O.K., Evdokimov, V.N. 1987. New technological processes and technical means for the production of dry feed of animal origin, developed by UkrNIImyasomolprom. (In Russian). Express information AgroNIITEIMMP, 53-54.
  2. Ivashov, V.I. 2001. Technological equipment of meat industry enterprises. Part 1. Equipment for slaughter and primary processing. (In Russian): Моscow. Коlоs, 418-432.

  1. In this review, more information is needed regarding the composting processes of animal waste. I suggest reviewing this book or similar literature:

"Sánchez, Mercedes et al. (2008). Animal Solid Waste Management through Composting Techniques. Closed Semi-Continuous Composters as a New Approach for in-Situ Carcasses Disposal."

Response 4. (Lines 1576-1581)

Thank you for your suggestion. This literature was helpful and we have used this literature.

Animal waste is widely used in feed preparation, production of bio seal, bio coal, bio-oil, glue, and fertilizer production [62].

  1. I suggest removing the term "As you know" throughout the document.

Response 5. Thank you for your suggestion. We have deleted this term all over the text.

  1. I suggest simplifying the conclusions, and focusing on the most relevant aspects found in the literature review.

Response 6. (Lines 1187-1418)

We have rewritten the conclusion. Please check!

Conclusion

Contemporary scientific publications and research are mainly devoted to the processing of insects into feed, the use of bird droppings in the production of biochar, bio-oil, and fertilizers, and also the preparation of feed additives. Meat and bone meal are mainly used for domestic animals and birds, and also find use, like bone meal, in heat supply. Research is being carried out extensively in the processing of fish waste, crustacean shells, shrimp production waste, crab by-products, bird waste, and eggshells. The existing and emerging waste of animal origin directs researchers to think in depth about the solution of the rational processing of this source of raw materials, to extract the most beneficial effect. At the same time, processing indicates the intensification and improvement of the quality of such important processes as drying, grinding, and mixing, which are necessary for the production of feed flour, feed mixtures, fertilizers, biofuels, and other useful products. One of the best processing methods is the combination of drying processes with grinding and mixing in one apparatus, which can significantly increase the rate of heat and mass transfer, and reduce the cost of producing useful products due to the simultaneous processing of waste in one apparatus. At the same time, it should be noted that the addition of the mixing process will make it possible to prepare feed from a variety of raw materials of animal origin, which will eliminate the sorting process, as well as reduce the duration of mixing due to the longitudinal-transverse, vortex movement of feed flour particles, which contributes to high uniformity.

Along with the fact that the combination of grinding-mixing with drying increases the uniformity of heat treatment and the distribution of product particles by fractions, and composition, and also allows for sufficiently deep dehydration of the material, there are some disadvantages, in particular, this is the uneven distribution of processed particles in the temperature flow, various moisture content in the processed particles of raw materials and in parts of the apparatus, which in some cases causes a certain difference in the moisture content in the dried particles of raw materials, and in some cases causes overheating of finely crushed particles along with simultaneous energy consumption and a decrease in the biological value of the product.

Dear reviewer, Thank you for your time and effort in reviewing our manuscript!

Round 2

Reviewer 1 Report

Dear Authors,

Thank you for the effort you put into revising the manuscript. I believe this version of the manuscript has a higher quality in comparison with the previous version and the research can contribute to the domain. Therefore, I recommend it for publication in its current version.

Good luck with your research.